

# Hydrological and lock operation conditions associated with paddlefish and bigheaded carp dam passage on a large and small scale in the Upper Mississippi River (Pools 14–18)

Dominique D. Turney[1,2], Andrea K. Fritts[3], Brent C. Knights[3], Jon M. Vallazza[3], Douglas S. Appel[3] and James T. Lamer[1]

[1] Illinois River Biological Station, Illinois Natural History Survey, University of Illinois, Havana, IL, United States of America
[2] Department of Biological Sciences, Western Illinois University, Macomb, IL, United States of America
[3] United States Geological Survey, Upper Midwest Environmental Sciences Center, La Crosse, WI, United States of America

Corresponding author
Dominique D. Turney,
ddturney@illinois.edu

## ABSTRACT

Movement and dispersal of migratory fish species is an important life-history characteristics that can be impeded by navigation dams. Although habitat fragmentation may be detrimental to native fish species, it might act as an effective and economical barrier for controlling the spread of invasive species in riverine systems. Various technologies have been proposed as potential fish deterrents at locks and dams to reduce bigheaded carp (i.e., silver carp and bighead carp (*Hypophthalmichthys* spp.)) range expansion in the Upper Mississippi River (UMR). Lock and Dam (LD) 15 is infrequently at open-river condition (spillway gates completely open; hydraulic head across the dam <0.4 m) and has been identified as a potential location for fish deterrent implementation. We used acoustic telemetry to evaluate paddlefish passage at UMR dams and to evaluate seasonal and diel movement of paddlefish and bigheaded carp relative to environmental conditions and lock operations at LD 15. We observed successful paddlefish passage at all dams, with the highest number of passages occurring at LDs 17 and 16. Paddlefish residency events in the downstream lock approach of LD 15 occurred more frequently and for longer durations than residency events of bigheaded carp. We documented upstream passages completed by two individual paddlefish through the lock chamber at LD 15, and a single bighead carp completed upstream passage through the lock chamber during two separate years of this study. We identified four bigheaded carp and 19 paddlefish that made upstream passages through the spillway gates at LD 15 during this study. The majority of the upstream passages through the spillway gates for both species occurred during open river conditions. When hydraulic head was approximately 1-m or greater, we observed these taxa opt for upstream passage through the lock chamber more often than the dam gates. In years with infrequent open-river condition, a deterrent placed in the downstream lock approach may assist in meeting the management goal of reducing upstream passage of bigheaded carps but could also potentially affect paddlefish residency and passage. Continued study to understand the effects of deterrents on native fish could be beneficial for implementing an integrated bigheaded carp control strategy. Understanding fish behavior at UMR dams is a critical

information need for river managers as they evaluate potential tools or technologies to control upstream expansion of bigheaded carp in the UMR.

## INTRODUCTION

The movement and dispersal of migratory fish species are important life-history characteristics that can be restricted by natural and artificial structures (*Kruk & Penczak, 2003*; *Zielinski, Voller & Sorensen, 2018*). Dams are known to impede fish passage in lotic systems (*Porto, McLaughlin & Noakes, 1999*; *Larinier, 2001*; *Knights et al., 2002*; *Zigler et al., 2004*). Hydraulic challenges (*e.g.*, velocity, turbulence) and structural impermeability can heavily impact upstream fish passage through dams and reduce connectivity between important feeding and spawning areas (*Northcote, 1998*; *Larinier, 2000*; *Zielinski, Voller & Sorensen, 2018*). Additionally, downstream migration through turbines or spillways may result in significant mortality of fish (*Larinier, 2002*; *Čada et al., 2006*). Barriers to passage can create potential implications for long-term population dynamics or in extreme cases, lead to the extirpation of a species or threaten biodiversity (*Larinier, 2000*; *Pess et al., 2008*; *Liermann et al., 2012*).

Although habitat fragmentation may be detrimental to native migratory fish species, impassable barriers can provide an effective, economical tool for controlling harmful and invasive species (*Rahel & McLaughlin, 2018*; *Altenritter et al., 2019*). Barriers that lead to fragmented systems can prevent the spread of nonnative species, exotic diseases, and hybridization (*Rahel, 2013*). Seasonally operated physical and electrical barriers have provided an effective management alternative to control sea lamprey (*Petromyzon marinus*) in the Laurentian Great Lakes, although they prohibit many non-jumping native fish to pass (*McLaughlin et al., 2007*; *Vélez-Espino et al., 2011*). Natural resource managers are faced with balancing the pros and cons of connectivity in aquatic systems (*Rahel, 2013*; *Rahel & McLaughlin, 2018*).

The upper Mississippi River (UMR) has been substantially modified over the past century with the construction of a series of 29 navigation locks and dams (LD). Each LD differs in design and the percentage of time in open-river condition, defined as the time when the adjustable spillway gates of the dam (*i.e.,* roller and tainter gates) are raised out of the water, passing unobstructed water through the gates (*Wilcox et al., 2004*). When the river is at open-river condition, the head and tail surface elevations of the river are nearly equal. Dams that experience frequent open-river condition likely support more upstream fish passage than those that have infrequent or no open-river condition because water velocity through the spillway is reduced compared to controlled conditions (*i.e.,* when partially lowered dam gates create accelerated water velocities and increased turbulence). For example, LDs 1, 2, 14, 15, and 19 are individually at open-river condition less than 2% of the time, or incapable of achieving this condition, likely impeding upstream fish

passage during most years (*Wilcox et al., 2004*). As such, LDs 14, 15, and 19, considered as pinch-point dams, have been identified as focal points for understanding the impact of infrequent open-river condition on native, non-native, and invasive fish passage.

The American paddlefish (*Polyodon spathula*) is a large-bodied, cartilaginous fish species endemic to the Mississippi River Basin, Gulf Coastal drainages, and historically in the Great Lakes (*Eddy & Underhill, 1978*; *Carlson & Bonislawksy, 1981*). Paddlefish were once an abundant species throughout the Mississippi River Basin, but overharvesting, habitat loss and fragmentation, and water pollution have resulted in population declines since the early 1900s (*Carlson & Bonislawksy, 1981*; *Sparrowe, 1986*; *Unkenholz, 1986*; *Graham, 1997*; *Jennings & Zigler, 2000*), leading to their classification as endangered, threatened, or a species of concern in several states (*Graham, 1997*). Paddlefish are highly migratory, capable of travelling great distances in short periods of time (*Rosen, Hales & Unkenholz, 1982*; *Southall, 1982*; *Russell, 1986*; *Tripp et al., 2019*) but their movement is restricted by the presence of navigation dams (*Larinier, 2000*; *Zigler et al., 2003*; *Zigler et al., 2004*). In the UMR, LDs have the potential to limit paddlefish movement and access to suitable habitats and spawning areas (*Zigler et al., 2004*). Also, populations of paddlefish might be further threatened by the presence of competing invasive fishes, such as bigheaded carp.

Silver carp (*Hypophthalmichthys molitrix*) and bighead carp (*H. nobilis*), hereafter referred to as bigheaded carp, are native to Eastern Asia and are highly invasive in the UMR (*Kolar et al., 2007*). These fishes have the capability to compete with native planktivores, such as paddlefish and gizzard shad (*Dorosoma cepedianum*), for food (*Schrank, Guy & Fairchild, 2003*; *Sampson, Chick & Pegg, 2009*). Bigheaded carp are highly mobile and since their introduction into the United States in the 1970s (*Koel, Irons & Ratcliff, 2000*; *Kolar et al., 2007*), their range and abundance has expanded, thus increasing their potential for causing ecological and economic damage (*Kolar et al., 2007*). Bigheaded carp can pass upstream through the gated portions of navigation dams (*Tripp et al., 2014*), as well as through lock chambers of navigation dams (*Lubejko et al., 2017*; *Fritts et al., 2021*). Understanding the environmental conditions and fish behaviors associated with dam passage is critical for informing controls to limit the upstream spread of bigheaded carp in the UMR (*Lubejko et al., 2017*).

UMR dams with attributes that make them more restrictive to upstream fish passage (*e.g.*, infrequent open-river condition, high vertical lift, challenging spillway design features) are considered focal points for fish passage and bigheaded carp management (*Wilcox et al., 2004*; *Upper Mississippi River Asian Carp Partnership, 2018*). Although there is potential for improving native fish passage at these pinch-points, these dams are also candidates for placement of deterrent technologies and gate manipulation to further limit the upstream movement of invasive bigheaded carp (*Zielinski, Voller & Sorensen, 2018*; *Finger et al., 2020*). Numerous behavioral deterrents are being designed and tested for restricting bigheaded carp movements including acoustic, carbon dioxide, electricity, and multi-sensory deterrents (*e.g.*, *Ruebush et al., 2012*; *Vetter et al., 2015*; *Cupp et al., 2016*; *Parker et al., 2016*; *Dennis, Zielinski & Sorensen, 2019*). Understanding the movement of native and nonnative migratory fishes through locks and dams prior to deterrent operation is important. For this reason, paddlefish and bigheaded carp were chosen as representative

species for study at strategic locks and dams in the UMR (*i.e.,* locks and dams near the invasion front for bigheaded carp) to inform future decisions regarding deterrents.

The objectives of this study were to better understand large- and small-scale movements and behaviors of paddlefish and bigheaded carp by (1) identifying and describing environmental factors that might be influencing the passage of paddlefish through LDs 14–18, (2) evaluating the effects of environmental variables and lock operation on the presence of bigheaded carp and paddlefish in the downstream approach of LD 15, (3) distinguishing diel and seasonal patterns of bigheaded carp and paddlefish residency in the downstream lock approach of LD 15, and (4) identifying the relations between fish passages through the LD 15 lock chamber with lock operations for paddlefish and bigheaded carp. This information can be used for modeling movement and inform the design and implementation of deterrent technologies to assist managers in restricting bigheaded carp movement while minimizing impacts on native fish passage through UMR LDs.

## MATERIALS & METHODS

### Study area

The study area included a 191-km reach from Pools 14 to 19 (Fig. 1). The study focused on three major pinch-point dams within this reach: LD 14 (river kilometer [rkm] 483), 15 (rkm 482), and 19 (rkm 364). Lock and Dam 14 is constructed of 13 tainter gates and four roller gates, a main lock, and an auxiliary lock for recreational vessels (*U.S. Army Corps of Engineers, 2018a*). Lock and Dam 15 is constructed of 11 roller gates and two locks, with a main and an auxiliary lock. There is also a hydropower dam located in a secondary channel on the east side of Arsenal Island. These dams are located approximately 2.4 km upstream of LD 15 on either side of Sylvan Island in Sylvan Slough (*U.S. Army Corps of Engineers, 2018b*). Locks and Dams 14 and 15 are infrequently at open-river condition and are only free-flowing between 1 and 2% of the year on average (*Wilcox et al., 2004*; *Bouska, 2021*). Lock and Dam 19 is a high-head hydroelectric dam and the dam gates have never been at open-river condition (*Wilcox et al., 2004*). Lock and Dam 19 is considered a major impediment to fish movement because all upstream passage is restricted to the lock chamber.

### Fish collection and surgery

One hundred twenty-one paddlefish were captured and tagged with acoustic transmitters in Pool 14 ($N = 59$) and Pool 16 ($N = 62$) during summer 2018. Paddlefish were captured using 13-cm mesh gill nets for both pools. Fish were weighed (g), measured (mm) as eye-to-fork length (EFL), and tagged with VEMCO (Nova Scotia, Canada) V16-6x acoustic transmitter tags (95-mm long, 34 g, 7-year battery life). The tag weight did not exceed 2% of the fish's total weight (*Winter, 1983*).

Surgical procedures performed are described in *Summerfelt & Smith (1990)* for the study. After the incision was closed, fish were transported to a recovery tank until full recovery (*i.e.,* equilibrium and normal swimming) was observed and the fish were released at the capture location. Fish collected for study were processed in accordance with the Institutional

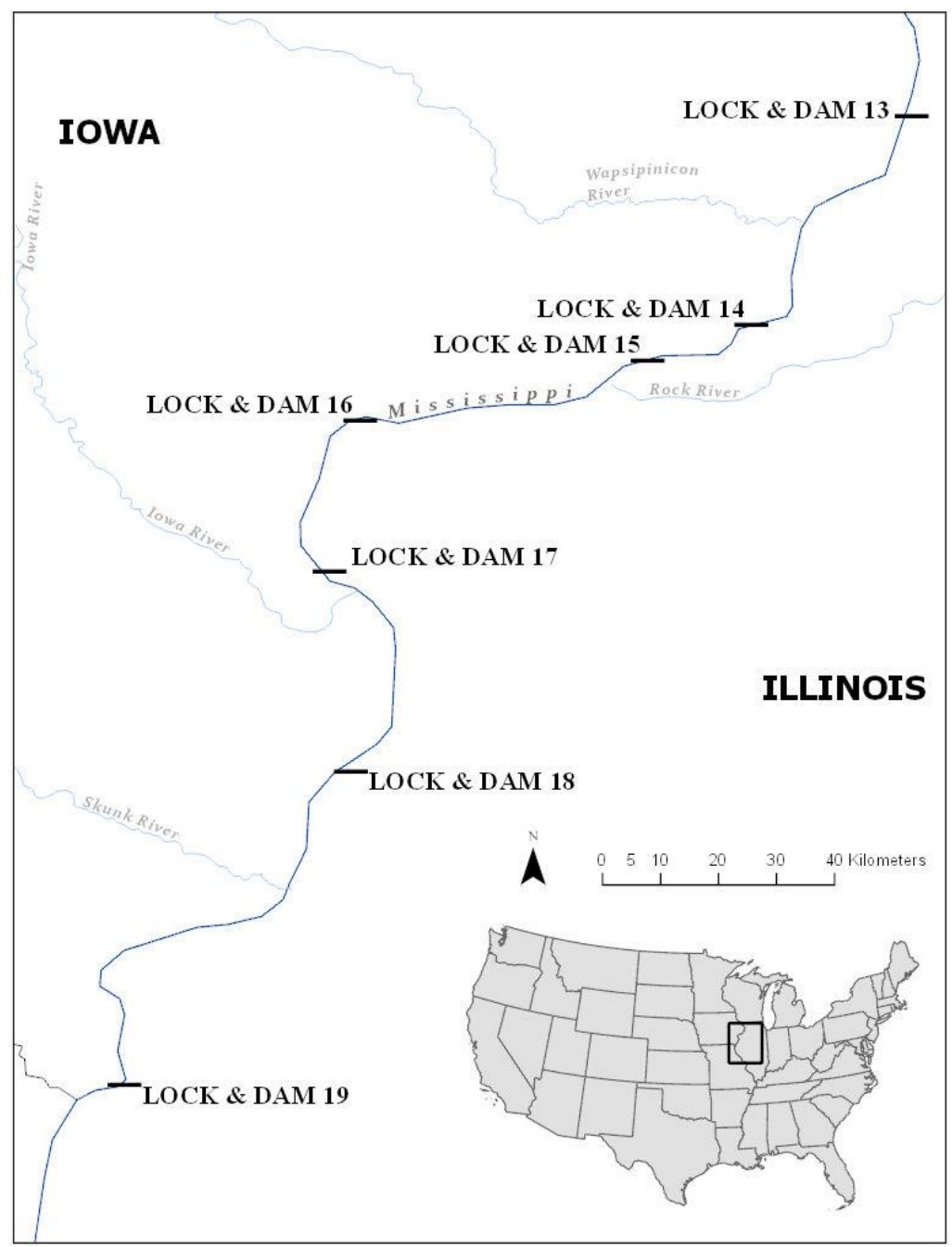

**Figure 1** Locations of locks and dams 14–19 on the upper mississippi river.

Animal Care and Use Committee (IACUC; IACUC Protocol 16-09 at Western Illinois University).

Bigheaded carp used for the study were previously captured with gill nets, tagged (VEMCO V16-6x) and deployed through a partnership with the U.S. Geological Survey (USGS) and the U.S. Fish and Wildlife Service (USFWS; *U.S. Fish and Wildlife Service, 2019*;

*U.S. Fish and Wildlife Service, 2020*). Captured bigheaded carp were handled in accordance with the U.S. Geological Survey Upper Midwest Environmental Sciences Center Animal Care and Use Committee approved procedures (Upper Midwest Environmental Sciences Center (UMESC) Animal Care and Use Committee protocol number GEN136.3 and ESB-18-ACBW-01).

## Stationary and manual tracking

An extensive array of stationary receivers (VEMCO VR2W, VR2C, VR2Tx) was used to monitor the movement of tagged fishes. Tagged fish were manually tracked with a mobile receiver (VEMCO Model VR100) and portable omni-directional hydrophone (VEMCO Model VH165; 50–85 kHz) to supplement the passive receiver array and obtain movement and habitat information of bigheaded carp and paddlefish. Manual tracking occurred weekly along a 0.5-km pre-determined grid within the study reach. At each waypoint, the omni-directional hydrophone was submerged for 100 s to detect presence of tagged fish. When a tag was identified, a VEMCO VH110 directional hydrophone (50–84 kHz) was used to obtain a more accurate location of the fish.

To improve understanding of passage by the focal species, a fine-scale receiver array composed of 15 stationary VR2Tx receivers was used (Fig. 2). There were 11 receivers in the downstream lock approach, two receivers inside the main lock chamber, and two receivers positioned above LD 15. Receivers were deployed within recessed ladder wells to protect the receivers and minimize collisions with vessels. Placing receivers in ladder wells resulted in some acoustic shadowing (*i.e.,* physical obstruction of sound waves), however testing conducted prior to the study verified receivers were able to detect transmitters throughout the downstream lock approach, lock chamber, and upstream lock approach. Range testing confirmed the ability to reliably detect transmitters upstream and downstream of the LD 15 lock approach, as well as within the LD 15 lock chamber.

*In-situ* range testing of the LD 15 fine-scale array receivers was conducted with an acoustic test tag (10-s transmission rate) of the same frequency and power of implanted fish transmitters. Range testing determines an approximate maximum range of detection and evaluates the detection efficiency within this range. At LD 15, three distinct zones were monitored to confirm the ability of the fine-scale array to track progression of transmittered fish moving through three lock zones: the downstream lock approach, the lock chamber, and the upstream lock approach. Range testing was performed by lowering the VEMCO test tag 1-m below the water's surface from a boat as it moved through the three lock zones. The path of the boat was recorded on a GPS unit, which logged a coordinate and timestamp at 1-s intervals. GPS coordinates and associated timestamps were then matched with the corresponding times of test tag detections on the receivers, resulting in the position of the test tag when it was detected. The time of first detection to the time of last detection in each zone was used to calculate the number of expected detections based on transmission rate—it was presumed the first and last detections defined the maximum detection ranges. Detection efficiency was the number of observed detections/number of expected detections × 100 (*Kim & Mandrak, 2016*). Detection efficiency is not directly related to the efficiency of detecting a fish in either of the three distinct zones. Because

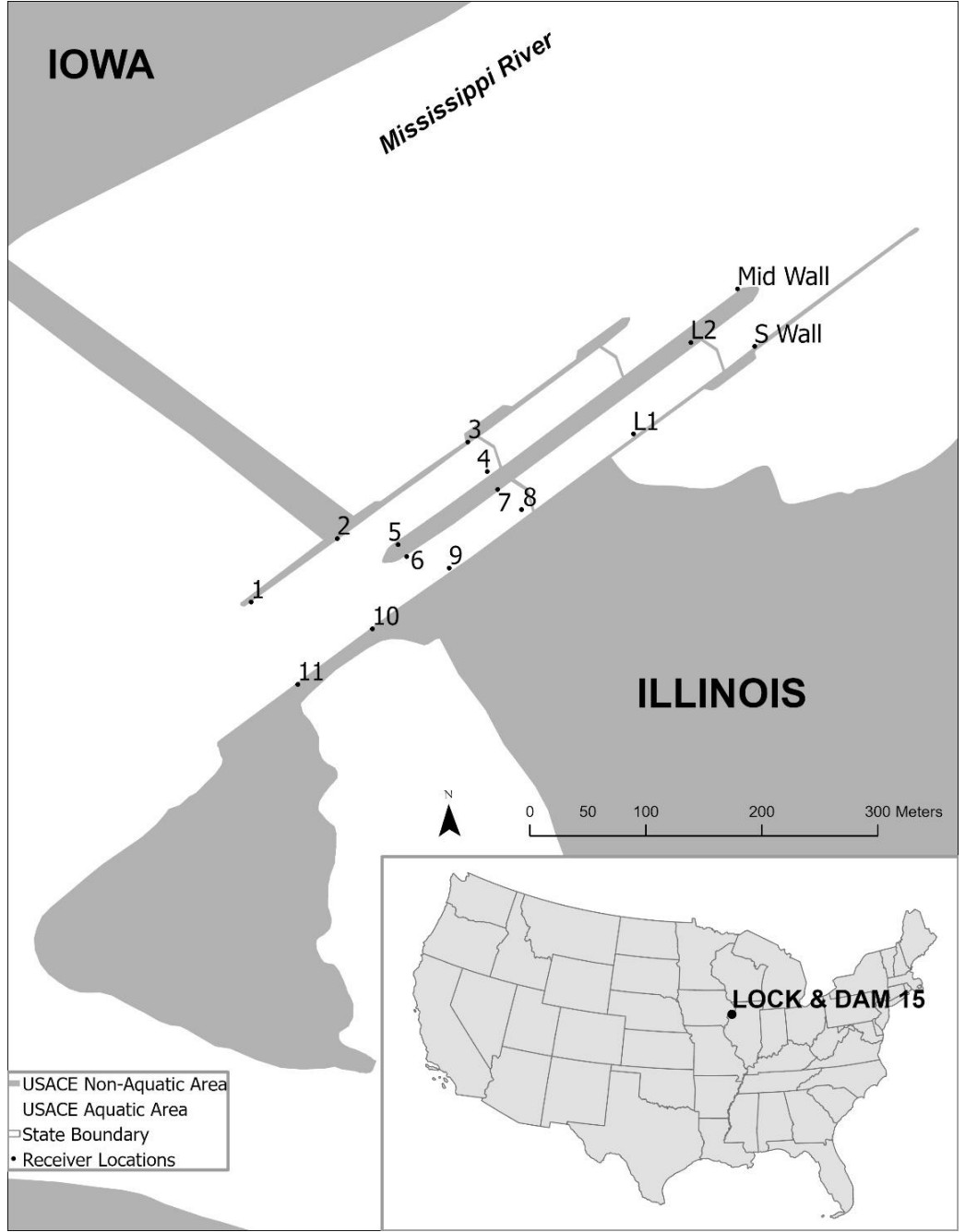

**Figure 2  Location of the study area at Lock and Dam 15 located in Davenport, Iowa, USA.** Fine-scale array receivers are denoted by the black dots. There were 11 receivers in the downstream lock approach (1–11), two receivers located inside the main lock (L1 and L2), and two receivers positioned in the upstream lock approach (Mid Wall and S Wall). Mid Wall is the receiver positioned above the upstream auxiliary lock approach. S Wall is the receiver positioned above the upstream main lock approach.

fish tags transmit a signal every 30–90 s, detecting only a few transmissions is necessary to confirm if a fish is present in a zone (*e.g.*, if efficiency were 50% with a nominal delay of 60 s, a fish detected in a zone for two minutes would be detected at least once).

## Statistical analysis

This study excluded individual detections of bigheaded carp and paddlefish during the first two weeks after surgical implantation of acoustic tags. This was done to minimize any altered fish behavior from surgical procedures and was established *a priori* (*Frank et al., 2009*; *Vallazza et al., 2021*). Daily water temperature and hydraulic head data were acquired from the U.S. Army Corps of Engineers (USACE), Rock Island District and lock operations and lockage event attributes from the USACE Lock Performance Monitoring System Lock Queue Report. Data generated during this study are publicly available through a USGS data release (*Fritts et al., 2022*; https://doi.org/10.5066/P9CHJ8OG). Due to the low number of individuals of silver carp ($n = 8$) and bighead carp ($n = 7$) below LD 15 and the biological similarities between the two, observations of bigheaded carp were combined for the analyses.

### Cox's proportional hazard regression for large-scale passage

Time-to-event (TTE) analysis (*Allison, 2014*; *Vallazza et al., 2021*) was used to quantify the hydraulic and environmental conditions associated with upstream and downstream passage by paddlefish at LDs 14–18. Use of the TTE model allows for: (1) inclusion of both time-dependent and time-independent covariates, (2) variable entrance and exit times of subjects (*e.g.*, due to variable tagging dates, fish mortality, tag loss, fish removal and study completion) and (3) repeated events (*e.g.*, a single fish moving past multiple dams). LD 19 was excluded from the analysis due to infrequent detections (*i.e.,* only a single downstream passage event). Findings from this paddlefish dam passage analysis were compared to results from a similar analysis of factors related to dam passage by bigheaded carps (*Vallazza et al., 2021*). Paddlefish that dispersed outside of LDs 14–18 were censored from analyses because observations outside of this boundary have unknown values for hydraulic and other unforeseen environmental conditions. We used a complimentary log–log model for continuous-time processes to approximate a Cox's proportional hazard model (*Allison, 2010*). Covariates modeled on the intensity of dam passage events (response variable) include daily mean water temperature (°C), daily mean hydraulic head (m), total fish length (mm), and sequence of passage events. Hydraulic head is the difference in height (m) between the river stage immediately upstream of the dam and the river stage immediately downstream of the dam. Sequence, in the context of a potentially repeated event (here, either consecutive upstream or downstream dam passages), refers to the order of the multiple dam passage events. A Pearson correlation measure was used to examine the relation between predictors to avoid multicollinearity. Predictors that had an $r > 0.50$ were not included in the analysis (*Dormann et al., 2013*).

Due to the rate of change of important covariates such as hydraulic head and water temperature, individual histories of tagged paddlefish were summarized as fish-days. Each fish-day was assigned a value of 0 (no dam passage observed) or 1 (dam passage observed).

The duration of dam passage events was defined as the date last detected in the first pool to the date first detected in the second pool and was typically >1 d, resulting in interval censored data. Therefore, covariate values for dam passages >1 d were summarized for the entire passage duration. For continuous variables (*e.g.*, hydraulic head and water temperature), the mean of the daily means was used for the passage interval. The resulting unequal time interval lengths this created was accommodated by treating time-from-previous-passage-event (t_from_dp) as a continuous variable and including a squared term (t_from_dp * t_from_dp) in the model to adjust for nonlinearity (*Allison, 2010*; *Vallazza et al., 2021*). Variation associated with individual fish behavior was accounted for by treating unique fish identity as a random effect in the model. Upstream and downstream dam passages were modeled separately. Akaike Information Criterion (AIC) values were used to compare the relative fit of all candidate models (*Akaike, 1973*). Confidence models were selected from models that had a ΔAIC <2 (*Royall, 1997*). The percentage of change of the hazard was calculated by subtracting one from the exponentiated coefficient estimates and multiplying by 100 (*Allison, 2010*). All TTE calculations were performed using SAS v.9.4 (*SAS Institute, 2012*).

### Residency and presence events at LD 15

Discrete residency events were calculated using the 'residence event' function in the VTrack package in R (*Campbell et al., 2012*; *R Core Team, 2019*). A residency event was defined by at least two detections in the downstream approach of the LD 15 fine-scale array within one hour. An event was considered "timed out" after an individual was not detected within one hour. Generalized linear mixed-effects models (GLMMs) were used to analyze bigheaded carp and paddlefish residency duration using the glmmTMB package in R (*Brooks et al., 2017*). Maximum likelihood estimation was used to compare models with their differing fixed effects (*Hosmer & Lemeshow, 2000*). A random effect was used for individual fish in the models because an individual could produce multiple residency events over the course of the study. The response variable was the duration of a residency event (minute). Using the DHARMa package in R, the residuals of the global GLMM were plotted to select the probability distribution that best fit the data (*Hartig, 2017*). A negative binomial distribution with a log link function was used for a suitable modeling distribution. All models were evaluated using AIC.

Presence events were examined as a binary response (*i.e.,* presence or absence of one or more individuals of paddlefish or bigheaded carp on a given day) in the downstream LD 15 approach. Presence event modeling allowed us to examine the effects of environmental variables and lock operations on a daily basis, as opposed to residency events that only allowed us to examine those data during days when fish were present (*Fritts et al., 2021*). The presence events were modeled using a generalized linear model (GLM) with a binomial distribution with a logit link function. Candidate models were compared using AIC (MuMin R package; *Akaike, 1973*; *Barton, 2019*). Presence of an individual (0 = not present or 1 = present) for a given day per paddlefish or bigheaded carp was chosen as the response variable for the candidate models. The presence data were split into an 80% model training dataset and a 20% test dataset. The predictive model performance was

**Table 1 Parameters and their hypothesized effects on residency and presence events.** Parameters are included in candidate models for bigheaded carp and paddlefish in the downstream lock approach of Lock and Dam 15 from 2017–2019. Bigheaded carp and paddlefish were modeled separately.

| Parameter | Interpretation |
|---|---|
| Water temperature (Temp) | Residency duration or presence may vary by temperature; temperature may serve as a spawning cue |
| Season | Residency duration or presence may change seasonally |
| Hydraulic head[a] (Hydraulic.head.m) | Residency duration or presence may vary by hydraulic head; hydraulic head may serve a spawning cue or initiate movement to low-flow refugia |
| Downstream-bound recreational vessel (n/day; Rec.D.n) | An increase in downstream-bound recreational vessels may decrease residency duration or presence of fish |
| Downstream-bound commercial tows (n/day; Barge.D.n) | An increase in downstream-bound commercial tows may decrease residency duration or presence of fish |
| Upstream-bound recreational vessel (n/day; Rec.U.n) | An increase in upstream-bound recreational vessels may increase residency duration or presence of fish |
| Upstream-bound commercial tows (n/day; Barge.U.n) | An increase in upstream-bound commercial tows may increase residency duration or presence of fish |
| Year | Residency duration or presence may change yearly; flood years could have an impact |

Notes.
[a] Hydraulic head is the difference in height (m) between the river stage immediately upstream of the dam and the river stage immediately downstream of the dam.

tested using the receiver operating characteristic curve (AUC) (ROCR R package; *Sing et al., 2005*). Cohen's kappa coefficient (*Cohen, 1960*; *Landis & Koch, 1977*) and the 20% test data set were used to evaluate the performance of the top model.

The explanatory variables in the residency and presence event models included average daily water temperature (°C), season (*Coulter et al., 2018*), year, hydraulic head (m), number of upstream-bound commercial tows per day, number of downstream-bound commercial tows per day, number of upstream-bound recreational vessels per day, and number of downstream-bound recreational vessels per day (Table 1). A Pearson correlation was used to examine multicollinearity between predictors. Predictors that had a $r > 0.50$ were not included in the analysis (*Dormann et al., 2013*).

In additional to the global model, 76 candidate models were created using combinations of the eight explanatory variables. Akaike Information Criterion (AIC) values were used to compare the relative fit of all candidate models (*Akaike, 1973*) and the best performing models were those which had the lowest AIC values. The best fitting candidate models displayed the highest model weights. From the candidate models, models that had a ΔAIC <2 were retained as the confidence set of models (*Royall, 1997*). The 95% confidence intervals (CI) of the estimate parameters were evaluated and if the CI overlapped zero, it was determined that the parameter was too imprecise to determine a relation (*Knol, Pestman & Grobbee, 2011*). The residual and normal probability plots were examined to assess the goodness-of-fit for the global model. The amount of variation explained by the best models was evaluated by calculating the coefficient of determination ($R^2$).

### Weekly and diel patterns at LD 15

The weekly and diel patterns of bigheaded carp and paddlefish presence were evaluated using the telemetry array at the downstream approach of LD 15. Weekly presence was calculated by the number of unique individuals (N) for paddlefish and bigheaded carp within each week during the study. Diel patterns of presence were calculated by the proportion of residency events within a given hour by each individual and then an average proportion of residency events was calculated for all individuals for bigheaded carp and paddlefish.

### Fine-scale LD 15 passage events

Upstream and downstream passages of bigheaded carp and paddlefish were identified using the LD 15 fine-scale telemetry array and the large-scale longitudinal telemetry array. Fish passage may occur either through the adjustable spillway gates (partially or fully opened), fixed-crest spillway (designed to release surplus flood water), or through the main and auxiliary lock chambers while boats are passing through these lock chambers. Both lock chamber gates remain closed when unused, making this portion of the dam impermeable to fish passage. The auxiliary lock is primarily used for smaller recreational vessels and is infrequently used compared to the main lock chamber.

The fish's position in the fine-scale telemetry array in the LD 15 downstream approach was used to determine the route of passage. If fish were detected in the LD 15 downstream lock approach, followed by a detection in the lock chamber (*i.e.,* L1 and L2; Fig. 2), then detected on receivers in the upstream lock approach (*i.e.,* S Wall or Mid Wall; Fig. 2), it was determined the fish passed upstream through the lock chamber. Downstream passage through the lock chamber was determined by a fish being detected on a receiver in the upstream lock approach, followed by a detection in the lock chamber, and then detected at a receiver in the downstream lock approach. Passages were presumed to have occurred through the dam gates of LD 15 if a fish was not detected in the fine-scale array or if a fish did not complete a sequence of detections that would indicate passage through the lock chamber. The USACE Lock Queue Report was used to determine the lock operations associated with the timing of fish passage through the lock chambers.

## RESULTS

The study was conducted from 01 Jan 2017 through 31 Dec 2019. During this period, water temperature ranged from −0.10 to 29.3 °C (mean = 13.0 °C) and hydraulic head ranged from −0.02 to 11.4 m (mean = 2.6 m) between Pools 14–19 (Table S1). During the study, open-river condition occurred at LD 14 7.0% of the time and at LD 15 12.5% of the time; LD 19 does not experience open-river condition (Table S1). Successful paddlefish passage was observed at all dams, with the greatest number of passages occurring at LD 17 (Table 2). At LD 15, where fine-scale passage with the receiver array was evaluated, there were a total of 14,318 tow and vessel lockages from 2017-2019 (Table S2).

Range testing at LD 15 occurred in April 2019 (Fig. S1) and December 2019 (Fig. S2). Testing in April occurred during major flooding and the average detection efficiency in the upstream lock approach, lock chamber, and downstream lock approach was 36%, 92%,

**Table 2 Upstream and downstream passage events of paddlefish through locks and dams (LD) on the Upper Mississippi River from 2017–2019.** Passages of paddlefish during the first two weeks after surgical implantation of acoustic tags in 2018 were excluded. No passage events by paddlefish were observed during 2017. The total is the combined number of upstream and downstream passages. N represents the number of unique individuals that completed upstream and downstream passages through the lock and dam. Fine-scale receiver arrays at LD 15 and LD 19 were used to identify the route of passage (*i.e.*, through the lock chamber or through the dam gates).

| Lock and Dam | Upstream | Downstream | Total | N |
|---|---|---|---|---|
| 14 | 19 | 3 | 22 | 19 |
| 15 | | | | |
|     Dam | 19 | 3 | 22 | 19 |
|     Lock | 2 | 0 | 2 | 2 |
| 16 | 10 | 15 | 25 | 14 |
| 17 | 17 | 25 | 42 | 14 |
| 18 | 5 | 9 | 14 | 5 |
| 19 | | | | |
|     Dam | 0 | 1 | 1 | 1 |
|     Lock | 0 | 0 | 0 | 0 |

and 89%, respectively. In December, the detection efficiency in the upstream lock approach increased considerably to 90%, while detection efficiencies in the other zones were similar to those observed on the previous test date (lock chamber and downstream lock approach were each 91%).

The number of unique individuals detected in the downstream approach at LD 15 varied weekly by paddlefish and bigheaded carp throughout the study period. Bigheaded carp were present in the lock approach during March through September, whereas paddlefish were present during March through November (Fig. 3). Bigheaded carp had the greatest number of residency events during the summer (77%; Table S3). The presence of paddlefish in the LD 15 downstream lock approach showed some seasonality as presence was highest during April and lower during June through November (Fig. 3). Paddlefish had the greatest number of residency events during the spring (55%) and summer (40%) months (Table S3). Bigheaded carp displayed diel patterns in the LD 15 downstream approach with greater proportions of residency events occurring from 02:00 to 09:00 CST (Fig. 4) while paddlefish did not exhibit any distinct diel patterns.

## Cox's proportional hazard

The relation between paddlefish dam passage at LDs 14–18 and the explanatory variables using a Cox's proportional hazards regression model included daily mean hydraulic head (m), daily mean water temperature (°C), sequence of passage, and EFL of paddlefish in the best approximating upstream passage model (Table 3). The most informative model (AIC = 831.88) for upstream dam passage by paddlefish indicated that the probability of passage decreased as hydraulic head increased ($p < 0.0001$) and increased as water temperature increased ($p < 0.0001$; Table 3). Based on model predictions, a 1-m increase in hydraulic head would result in a 59%–84% (95% CI) decrease in the rate of upstream passage of paddlefish. The rate of upstream passage would increase 7%–16% with each 1 °C increase

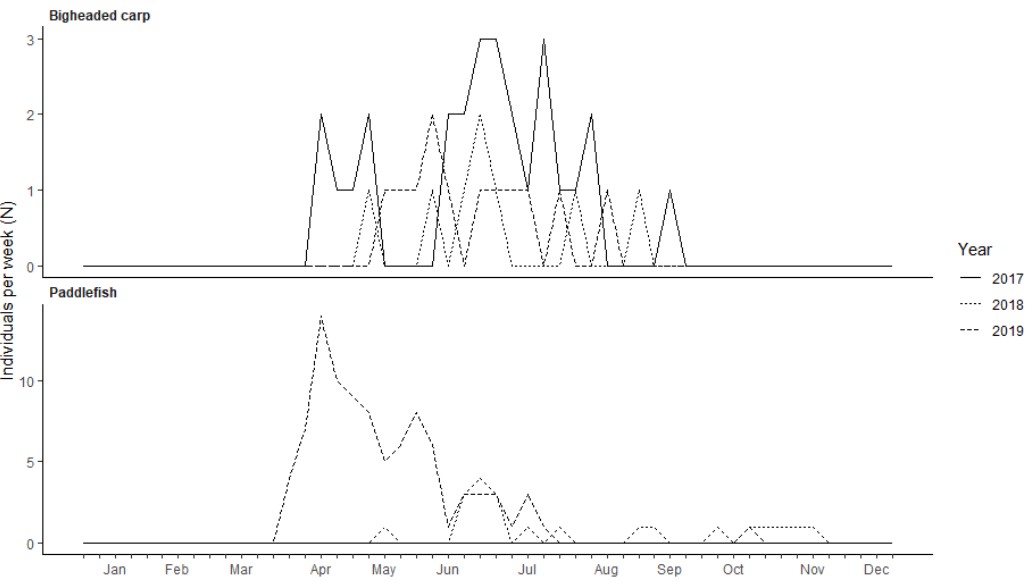

**Figure 3 Number of individuals (N) per week present below Lock and Dam (LD) 15 during January 2017 through December 2019.** Years include 2017 (solid line), 2018 (dotted line), and 2019 (dashed line). There was a total of 15 bigheaded carp and 43 paddlefish detected below LD 15 during the study. Note different scales on the y-axes.

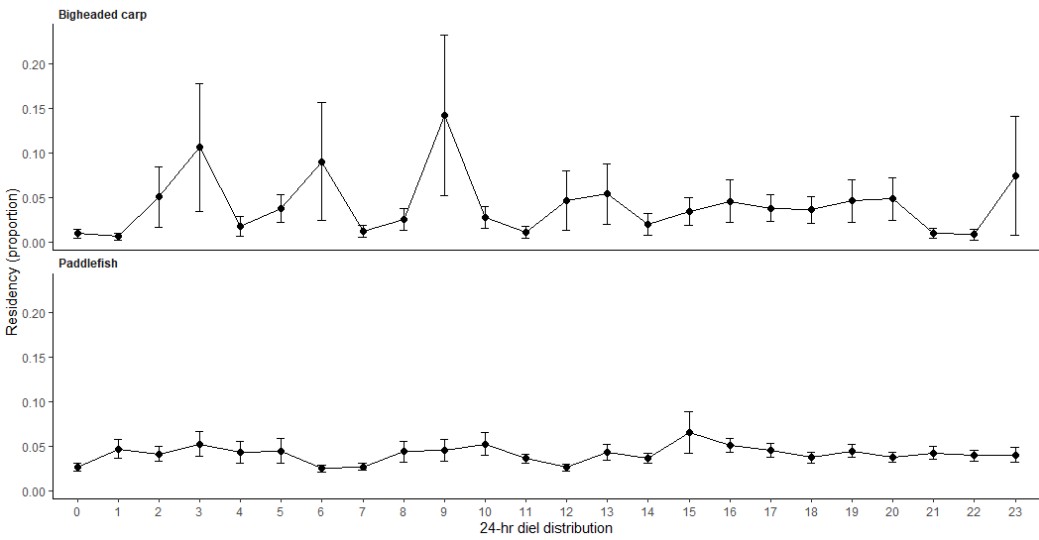

**Figure 4 Diel patterns of fish presence in the downstream lock approach at Lock and Dam (LD) 15 during January 2017 through December 2019.** Diel patterns were calculated using the proportion of residence events within a given hour by each individual, then averaging these proportions of residence events (± SE) for all individuals for bigheaded carp and paddlefish. Individual bigheaded carp and paddlefish present at the downstream LD 15 lock approach during the study were 15 and 43, respectively.

**Table 3   Cox proportional hazards models with Akaike's information criterion (AIC) and AIC of upstream and downstream passages of paddlefish at Locks and Dams 14–18 in the upper Mississippi River.** The 95% confidence interval (CI) of the hazard ratio (upper = UCI; lower = LCI) is calculated by subtracting one from the exponentiated CI estimate and multiplying by 100. These values are reported as percentages.

| Model | AIC | ΔAIC | Variable | Estimate | SE | p-value | UCI | LCI |
|---|---|---|---|---|---|---|---|---|
| **Upstream** | | | | | | | | |
| head[a]*temp[b]*seq[c]*length[d] | 831.88 | 0 | head | −1.36 | 0.23 | <0.0001 | −0.91 | −1.82 |
| | | | temp | 0.11 | 0.02 | <0.0001 | 0.15 | 0.07 |
| | | | seq | 0.24 | 0.14 | 0.09 | 0.53 | −0.04 |
| | | | length | −0.01 | 0.003 | 0.09 | 0.001 | −0.01 |
| head*temp*length | 832.66 | 0.78 | head | −1.40 | 0.23 | <0.0001 | −0.94 | −1.86 |
| | | | temp | 0.11 | 0.02 | <0.0001 | 0.16 | 0.07 |
| | | | length | −0.01 | 0.003 | 0.08 | 0.001 | −0.01 |
| head*temp*seq | 832.86 | 0.98 | head | −1.36 | 0.23 | <0.0001 | −0.90 | −1.81 |
| | | | temp | 0.11 | 0.02 | <0.0001 | 0.15 | 0.07 |
| | | | seq | 0.24 | 0.15 | 0.09 | 0.53 | −0.04 |
| head*temp | 833.67 | 1.79 | head | −1.39 | 0.23 | <0.0001 | −0.94 | −1.85 |
| | | | temp | 0.11 | 0.003 | <0.0001 | 0.16 | 0.07 |
| null | 914.51 | 82.63 | | | | | | |
| | | | | | | | | |
| **Downstream** | | | | | | | | |
| head*temp*seq | 609.06 | 0 | head | −0.71 | 0.296 | 0.02 | −0.14 | −1.30 |
| | | | temp | 0.06 | 0.022 | 0.01 | 0.10 | 0.02 |
| | | | seq | −0.38 | 0.148 | 0.01 | −0.09 | −0.67 |
| head*temp*seq*length | 609.30 | 0.24 | head | −0.66 | 0.283 | 0.02 | −0.11 | −1.22 |
| | | | temp | 0.06 | 0.022 | 0.01 | 0.10 | 0.01 |
| | | | seq | −0.40 | 0.150 | 0.01 | −0.11 | −0.69 |
| | | | length | −0.01 | 0.005 | 0.24 | 0.004 | −0.02 |
| temp*seq | 610.58 | 1.52 | temp | 0.04 | 0.021 | 0.04 | 0.08 | 0.002 |
| | | | seq | −0.48 | 0.145 | 0.001 | −0.20 | −0.77 |
| null | 611.93 | 2.87 | | | | | | |

**Notes.**
[a] Hydraulic head is the difference in height (m) between the river stage immediately upstream of the dam and the river stage immediately downstream of the dam.
[b] Water temperature recorded at the dam of passage.
[c] Sequence refers to the order of the consecutive upstream or downstream dam passages by an individual.
[d] Eye-to-fork length of an individual.

in water temperature. The effects of sequence and EFL on upstream dam passage were not significant ($\alpha \geq 0.05$). All four models in the confidence set included hydraulic head and water temperature.

The best approximating downstream model included hydraulic head, daily mean water temperature, and sequence of passage covariates. The most informative model for downstream dam passage by paddlefish (AIC = 609.06) indicated that the probability of passage decreased as hydraulic head increased ($p = 0.016$), increased as temperature increased ($p = 0.01$), and decreased as sequence increased ($p = 0.01$; Table 3). Each 1-m increase in hydraulic head would result in an expected 13–73% decrease in the probability of a downstream dam passage. A 1 °C increase in water temperature would result in an expected 2–11% increase in the probability of a downstream dam passage. The

**Table 4 Summary statistics for residency events for bigheaded carp and paddlefish in the downstream lock approach at Lock and Dam 15 from 2017–2019.** Individuals (N) is the number of unique individuals detected and RE is the number of unique residency events. Summary statistics included mean (minutes) and (±) standard error of residency duration, median, minimum (min), and maximum (max) of residency duration (minutes).

| Species | Individuals (N) | RE | Mean | Median | Min | Max |
|---|---|---|---|---|---|---|
| Bigheaded carp | 15 | 133 | 84 ± 11 | 32 | 1 | 734 |
| Paddlefish | 42 | 533 | 176 ± 19 | 64 | 1 | 6414 |

probability of occurrence of an additional downstream dam passage would decrease with each successive downstream dam passage. All three models in the confidence set included water temperature and sequence.

## Residency duration

Residency duration in the LD 15 downstream lock approach varied among paddlefish and bigheaded carp during the study period. We observed 133 bigheaded carp residency events from 15 individuals and 533 paddlefish residency events from 42 individuals (Table 4). The median residency duration for bigheaded carp and paddlefish was 32 min and 64 min, respectively. For paddlefish, 52% of the observed residency events were >1-hr (Table 4). Residency duration for bigheaded carp ranged from one min–12 h and paddlefish residency duration ranged from one min–100 h (Table 4).

The GLMM evaluated the relation between residency duration of bigheaded carp and paddlefish in the downstream lock approach of LD 15 and different environmental and lock operation parameters (Table 1). Bigheaded carp and paddlefish were evaluated separately as the biological differences between paddlefish and bigheaded carp were hypothesized to impact their response to the predictors (*Fritts et al., 2021*). Therefore, the results for the residency event GLMMs have been presented separately for paddlefish and bigheaded carp.

There were seven models in the residency duration confidence set for bigheaded carp (Table S4). The most informative bigheaded carp model included three parameters: water temperature, season, and the number of recreational vessels moving downstream (Table S4). The most informative model indicated that an increase in the number of downstream-bound recreational vessels decreased bigheaded carp residency duration (95% CI [−0.33−−0.03]) at LD 15 (Table 5). Additionally, bigheaded carp exhibited longer residency durations at LD 15 in the spring (95% CI [0.57–4.11]) and summer (95% CI [0.25–3.31]) relative to the fall (Table 5). The CI for water temperature overlapped zero and was considered too imprecise to establish a relation with residency duration (Table 5).

The most informative paddlefish residency duration model included five parameters: water temperature, number of commercial and recreational vessels moving downstream, and number of commercial and recreational vessels moving upstream (AIC = 6320.1; Table S4). There were three models in the confidence set for paddlefish (Table S4) and the $R^2$ for the best model was 12% (Table 5). The most informative model suggested an increase in the number of commercial tows moving downstream (95% CI [0.02–0.12]) and recreational vessels moving upstream (95% CI [0.05–0.52]) would increase the duration of paddlefish residency at LD 15 (Table 5). An increase in water temperature (95% CI [−0.08−−0.03])

**Table 5  Parameter estimates for the best supported residency duration generalized linear mixed-effects models for bigheaded carp and paddlefish in the downstream approach at Lock and Dam 15 from 2017–2019.** Standard error (in parentheses) and the upper and lower 95% confidence interval are included for each estimate. $R^2$ represents the coefficient of determination. Definitions of each parameter are located in Table 1.

| Parameter | Estimate | Lower | Upper |
|---|---|---|---|
| Bigheaded carp, $R^2 = 38\%$ | | | |
| Intercept | −0.49 (1.50) | −3.43 | 2.45 |
| Temp | 0.10 (0.05) | −0.01 | 0.21 |
| Rec.D.n | −0.18 (0.08) | −0.33 | −0.03 |
| Season, Spring (in relation to fall) | 2.34 (0.90) | 0.57 | 4.11 |
| Season, Summer (in relation to fall) | 1.78 (0.78) | 0.25 | 3.31 |
| Paddlefish, $R^2 = 12\%$ | | | |
| Intercept | 5.59 (0.24) | 5.13 | 6.06 |
| Temp | −0.06 (0.01) | −0.08 | −0.03 |
| Barge.D.n | 0.07 (0.02) | 0.02 | 0.12 |
| Barge.U.n | 0.002 (0.02) | −0.04 | 0.04 |
| Rec.D.n | −0.32 (0.11) | −0.55 | −0.10 |
| Rec.U.n | 0.29 (0.12) | 0.05 | 0.52 |

and the number of recreational vessels moving downstream (95% CI [−0.55–−0.10]) suggested a decrease in paddlefish residency duration at LD 15 (Table 5). The parameter for the number of commercial tows moving upstream had a CI that overlapped zero and was considered too imprecise to establish a relation with residency duration (Table 5).

## Presence events

The presence of bigheaded carp and paddlefish in the downstream approach of LD 15 varied by paddlefish and bigheaded carp throughout the study. Bigheaded carp were present in the LD 15 downstream approach for 74 days from April to September, primarily in June and July (69%) (Table S5). Most bigheaded carp presence events occurred in 2017, followed by 2018, and then 2019. Paddlefish were present in the LD 15 downstream approach for 137 days from April to November (Table S5). The presence event GLM was similar between bigheaded carp and paddlefish in that water temperature, year, number of upstream-bound tows, number of upstream-bound recreational vessels, and hydraulic head were all included in the best models (Table S6). Water temperature, hydraulic head, and number of upstream-bound recreational vessels were included in all models in the confidence set for both paddlefish and bigheaded carp (Table S6).

The most informative presence event GLM for bigheaded carp (AIC $= 312.1$, $\kappa = 0.43$; Table S6) indicated that as water temperature (95% CI [0.18–0.34]) and the number of recreational vessels moving upstream (95% CI [0.02–0.53]) increased, the probability of bigheaded carp presence below LD 15 increased (Table 6). As hydraulic head (95% CI [−1.28–−0.55]) increased, the probability of bigheaded carp presence below LD 15 decreased. The probability of bigheaded carp presence in the LD 15 downstream lock approach decreased over the course of the study (Table 6). The number of commercial
**Table 6** **Parameter estimates for the best supported presence event generalized linear model for bigheaded carp and paddlefish in the downstream approach at Lock and Dam 15 from 2017–2019.** Standard errors (in parentheses) and the upper and lower 95% confidence interval are included for each estimate. Definitions of each parameter are located in Table 1.

| Parameter | Estimate | Lower | Upper |
|---|---|---|---|
| Bigheaded carp | | | |
| Intercept | −4.71 (0.76) | −6.33 | −3.32 |
| Temp | 0.26 (0.04) | 0.18 | 0.34 |
| Hydraulic.head.m | −0.90 (0.18) | −1.28 | −0.55 |
| Barge.U.n | −0.08 (0.05) | −0.17 | 0.01 |
| Rec.U.n | 0.28 (0.13) | 0.02 | 0.53 |
| Year 2018 (in relation to 2017) | −2.19 (0.47) | −3.18 | −1.31 |
| Year 2019 (in relation to 2017) | −1.41 (0.40) | −2.21 | −0.64 |
| Paddlefish | | | |
| Intercept | −19.28 (890.14) | −362.44 | 9.66 |
| Temp | 0.02 (0.03) | −0.04 | 0.09 |
| Hydraulic.head.m | −1.11 (0.18) | −1.48 | −0.77 |
| Barge.D.n | −0.11 (0.05) | −0.21 | −0.02 |
| Barge.U.n | 0.01 (0.04) | −0.07 | 0.09 |
| Rec.D.n | −0.06 (0.16) | −0.40 | 0.23 |
| Rec.U.n | 0.56 (0.19) | −0.33 | 0.42 |
| Year 2018 | 18.18 (890.14) | −12.3 | 352.85 |
| Year 2019 | 18.21 (890.14) | −10.99 | 359.94 |
| Season, Spring (in relation to fall) | 1.46 (0.44) | 0.63 | 2.37 |
| Season, Summer (in relation to fall) | 2.51 (0.58) | 1.43 | 3.71 |
| Season, Winter (in relation to fall) | −15.63 (1062.10) | −409.59 | 21.72 |

tows moving upstream had CIs that overlapped zero and was considered too imprecise to determine a relation.

The most informative GLM for paddlefish presence in the LD15 downstream lock approach was the global model (AIC = 372.5, $\kappa = 0.64$; Table S6) and indicated that an increase in hydraulic head (95% CI [−1.48–−0.77]) and number of commercial tows moving downstream (95% CI [−0.21–−0.02]) decreased the probability of paddlefish presence below LD 15 (Table 6). Paddlefish had a higher probability of being present at LD 15 in the spring (95% CI [0.63–2.37]) and summer (95% CI [1.43–3.71]) relative to fall (Table 6). Water temperature, year, the number of commercial and recreational vessels moving upstream, and the number of recreational vessels moving downstream had CIs that overlapped zero and were considered too imprecise to determine a relation.

## Passage events at LD 15

During the three years of the study, successful upstream and downstream passages of bigheaded carp and paddlefish have been identified through LD 15 using the fine-scale telemetry array receivers and the large-scale longitudinal array. Passages outside of the

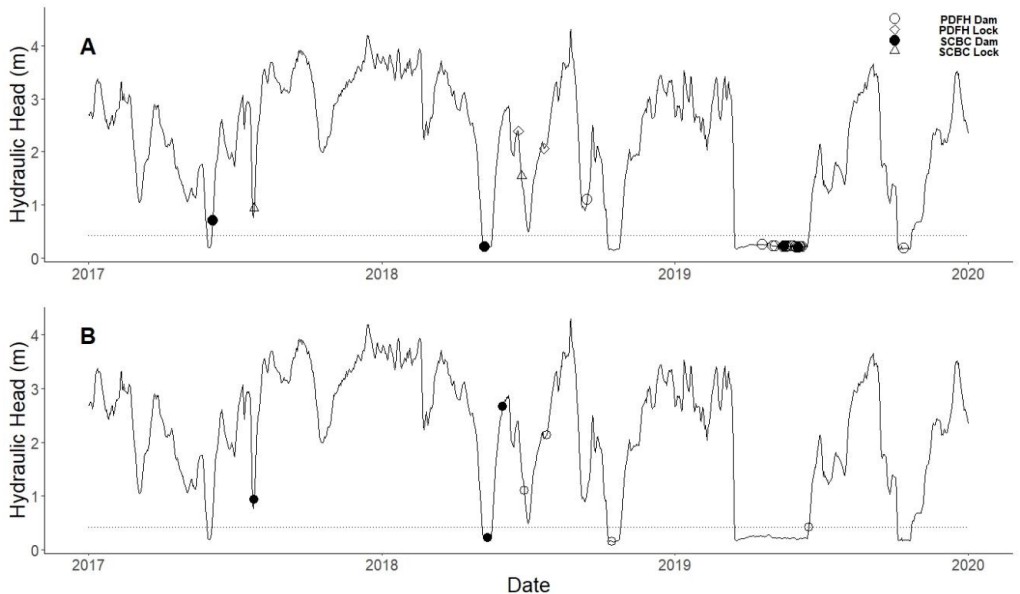

**Figure 5  Successful upstream and downstream bigheaded carp and paddlefish passages through the dam gates and lock chamber at Lock and Dam (LD) 15 from 2017–2019.** Upstream passage (A) and downstream passage (B) is plotted with the average hydraulic head (m). Triangles represent bigheaded carp passages through the lock, diamonds represent paddlefish passages through the lock, filled circles represent bigheaded carp passages through the dam gates, and open circles represent paddlefish or approaching passages through the dam gates. The horizontal dotted line represents the hydraulic head when LD 15 is at open-river conditions. There were four bigheaded carp and 20 paddlefish upstream passages through the dam gates and two bigheaded carp and two paddlefish upstream passages through the lock chamber (A). There were three bigheaded carp and four paddlefish downstream passages through the dam gates (B).

fine-scale array were presumed to have occurred through the adjustable spillway gates of LD 15. Bigheaded carp had a total of six upstream passages and three downstream passages completed by four individuals (three silver carp and one bighead carp; Fig. 5). All downstream passages occurred through the adjustable spillway gates. Two bigheaded carp upstream passages occurred through the lock chamber and four upstream passages were presumed to have occurred through the adjustable spillway gates. The two upstream passages through the lock chamber were made by one individual (bighead carp, female) over two years, 2017 and 2018. The first passage occurred on 26 July 2017 when this individual entered the lock chamber before the entrance of an upstream-bound commercial tow with six empty barges and exited with the same tow. The second upstream passage occurred on 24 June 2018 when the same individual entered the lock chamber before the entrance of an upstream-bound recreational vessel and exited with the same vessel. The hydraulic head at the dam during the first and second passages through the lock chamber were 0.93 and 1.55 m (*i.e.,* controlled conditions), respectively. This same individual made a third passage upstream in 2019, presumably through the adjustable spillway gates between June and July 2019 while the river was above flood stage and the dam was at open-river condition.

A total of 22 upstream passages and four downstream passages through LD 15 were completed by 21 individual paddlefish (Fig. 5; Table 2). All downstream passages occurred through the dam gates. Two paddlefish upstream passages occurred through the lock chamber and 20 of the upstream passages were presumed to have occurred through the adjustable spillway gates. The upstream passages through the lock chamber were made by two individuals in 2018. The first passage occurred on 20 June 2018, entering the lock chamber with an upstream-bound commercial tow consisting of 15 empty barges and one loaded barge. The fish entered the lock chamber with the first portion of the nine empty barges and exited upstream with the same tow. The second passage occurred on 22 July 2018 and the individual entered the lock chamber with an upstream-bound commercial tow consisting of 16 empty barges and exited upstream with the first cut of nine empty barges. The hydraulic head during these passages were 2.39 and 2.06 m (*i.e.,* controlled condition), respectively.

Nearly all observed upstream passages for bigheaded carp and paddlefish through the adjustable spillway gates occurred during open-river condition when hydraulic head was <0.4 m (22 of 24; Fig. 5). The two passages through the adjustable spillway gates that occurred outside open-river conditions were completed by a bigheaded carp on 13 June 2017 and a paddlefish on 14 September 2018, when hydraulic head was 0.70 m and 1.10 m, respectively. When hydraulic head was approximately 1-m or greater, paddlefish and bigheaded carp opted for upstream passage through the lock chamber more often than the dam gates (five of seven passages). Both bigheaded carp and paddlefish were able to achieve downstream passages during periods when hydraulic head was >1.08 m ($\pm$ 0.97 m).

## DISCUSSION

This study provided large- and fine-scale evaluations of invasive and native fish behaviors including passage at UMR dams. Locks and Dams 14, 15, and 19 are focal locations for fish passage studies as they are three of the most restrictive dams for upstream fish passage in the UMR (*Wilcox et al., 2004*). These locks and dams have been identified as potential locations for fish deterrent technologies to limit bigheaded carp range expansion in the UMR (*Upper Mississippi River Asian Carp Partnership, 2018*). Fish passage information for bigheaded carp and native fish species at these dams could be useful for assessing the potential ramifications of a deterrent on both groups. The results of this study advance the current understanding of bigheaded carp and paddlefish passage frequency and timing and how it is related to environmental conditions and lock operations at UMR dams.

Cox's proportional hazards model for paddlefish dam passage showed differing results for the upstream and downstream models. Water temperature and hydraulic head were important factors to upstream fish passage through Locks and Dams 14–18. The majority (68%) of paddlefish upstream passages occurred in the spring and early summer months. Water temperatures during these months possibly cued spawning behaviors of paddlefish while the low hydraulic head and associated lower current velocities through the adjustable spillway gates at open-river likely offered less resistance to passage for this species compared to controlled conditions (*i.e.,* periods with dam gates lowered) with high hydraulic head.

When water temperatures reach 10 °C, paddlefish begin to congregate in deep pools and start moving upstream in the river in search of inundated gravel bars to spawn (*Russell, 1986*). In addition to water temperature, increased water velocities and turbulence resulting from partially closed dam gates may exceed the swimming performance of paddlefish, resulting in a decreased presence of paddlefish during periods of high hydraulic head. Velocities through the UMR gates have been estimated as low as 0.6 m/s from a physical model study (*Markussen & Wilhelms, 1987*; *Wilcox, 1999*). *Zigler et al. (2004)* found that when hydraulic head was low (<1.0 m), there was a greater opportunity for upstream passage of paddlefish. Although adult paddlefish have morphological differences from other fish species that increase their critical swimming speed (*i.e.,* 0.86 m/s; *Wilcox et al., 2004*), high hydraulic head could increase stress, energetic costs, and injury (*Haro et al., 2004*). Downstream passage is likely more easily achieved by paddlefish than upstream passage over these dams (*i.e.,* swimming with the current is less energetically costly), which could explain the absence of significant factors in the regression model.

Bigheaded carp passage similarly is related to water temperature and hydraulic head. Water temperatures likely cue spawning behaviors in the UMR (*Vallazza et al., 2021*), and previous studies have shown that most upstream passages of bigheaded carp occurred when water temperature was ≥ 17 °C (*Larson et al., 2017*), and during open-river condition when the hydraulic head was <0.2 m (*Tripp et al., 2014*; *Lubejko et al., 2017*). Bighead and silver carp are capable of sustained swimming speeds of >0.98 m/s for >10 min (*Hoover, Zielinski & Sorensen, 2017*), which is greater than the lowest velocities reported through UMR dam gates (*Wilcox, 1999*). Yet, bigheaded carp approaching more restrictive dams, like LD 15, may be more likely to use the lock chamber to avoid high current velocities and turbulence below partially closed gates (*Vallazza et al., 2021*). Downstream passage was not affected by the same constraints as upstream passages as bigheaded carp were able to make downstream migrations during less favorable passage conditions (*Vallazza et al., 2021*).

Fish passage through the LD 15 lock chamber has been observed to coincide with vessel lockage in this study. Three upstream passages coincided with the upstream-bound lockage of commercial tows and one passage occurred with an upstream-bound recreational vessel. Our results revealed some important differences from a previous study of UMR dam passages *via* the lock chamber (*Fritts et al., 2021*). Similar to our results, the authors documented nearly all bigheaded carp and paddlefish initiating upstream passage *via* the lock chamber coinciding with an upstream lockage of a commercial tow. Unlike the previous study, we documented fish entrance and exit of the lock chamber occurring during the same vessel lockage event, compared to the previously-described multiple vessel lockages required to complete fish passage. Also, we observed fish passage *via* the lock chamber occurring in conjunction with a recreational vessel lockage, which is novel to our study. Additional observations of passage through the lock chamber are needed to establish a more definitive relation between lock operations, hydraulic conditions, and lock chamber passage of bigheaded carp and paddlefish at LD 15.

The residency duration and presence events in the downstream lock approach at LD 15 differed greatly between bigheaded carp and paddlefish. The longer residency events of paddlefish compared to bigheaded carp are similar to findings by *Fritts et al. (2021)* at the

downstream lock approach of LD 19. Bigheaded carp were present in the lock approach at LD 15 between May–September, whereas paddlefish showed seasonal patterns preferring spring months (April–May). At LD 25 on the UMR, seasonal fish densities in the lock chamber found a similar occurrence of high fish densities occurring during the spring, summer, and fall months and having lower fish densities in the winter (*Johnson et al., 2005*; *Keevin et al., 2005*). The ability for bigheaded carp to spawn within a wide range of water temperatures (18–30 °C; *Kolar et al., 2007*) and exhibit protracted spawning (*Schrank & Guy, 2002*; *Camacho et al., 2020*), may cause bigheaded carp to have longer presence in the lock approach as they are able to take advantage of optimal increases in discharge levels even after spring peak discharges (*Vallazza et al., 2021*). The seasonal differences between presence of bigheaded carp and paddlefish could be exploited by managers, as a deterrent in the lock approach might be most effective for managing bigheaded carp, with minimal impacts to paddlefish, if used during the mid-summer months.

Bigheaded carp residency duration was influenced by the number of recreational vessels moving downstream, water temperature, and season. Propeller strikes, noises, and bubbles from recreational vessels have been shown to increase stress levels in fish, provoking a "flight response" that leads to displacement of fish away from passing boats (*Becker et al., 2013*). Bigheaded carp residency duration also increased in the spring and summer, relative to the fall, coinciding with months in which water temperatures are suitable to bigheaded carp spawning. Paddlefish residency duration was influenced by water temperature, the number of commercial tows and recreational vessels moving downstream and the number of commercial tows and recreational vessels moving upstream. Studies have shown that in addition to photoperiod and water flow, water temperature is an important spawning cue for paddlefish (*Russell, 1986*). The optimum range for paddlefish spawning is 10 °C–20 °C, which occurred during April through June in the study (*Purkett Jr, 1961*; *Hubert et al., 1984*; *Wallus, 1986*). As water temperatures increase beyond this range, upstream migrations through dams would likely decrease.

The most important factors for paddlefish presence events were water temperature, hydraulic head, number of commercial tows and recreational vessels moving downstream, number of commercial tows and recreational vessels moving upstream, year, and season. A decrease in paddlefish presence due to downstream-bound commercial tows could be linked to several attributes of the movement and construction of tows. The shear forces, wake, and currents created by a commercial tow operation have been found to have direct and indirect impact on fish assemblages that can lead to injury or mortality of fish (*Wolter & Arlinghaus, 2003*). Loaded downstream-bound commercial tows have been documented on average to pass 228% of the water volume of the lock through the wheels (*i.e.*, tug propellers), compared to 49% passed on average with unloaded upstream-bound commercial tow (*Maynord, 2005*). This may cause additional displacement or mortality to fish that enter a lock chamber with a downstream-bound commercial tow leading to avoidance. *Barry et al. (2007)* documented paddlefish having a strong avoidance for the frequency emitted by commercial tows, finding that paddlefish immediately fled from an approaching tow and did not return until the tow was more than 2-km from the point of interaction. The specialized inner ear ultrastructure of paddlefish might be damaged by

the sonic emissions from tows that could elicit an avoidance response (*Lovell et al., 2006*). Furthermore, *Gurgens, Russell & Wilkens (2000)* found that the highly developed rostrum of paddlefish can detect and avoid metal objects, suggesting that the large metallic structure of tow hulls and miter gates might elicit an avoidance behavior by paddlefish.

## CONCLUSION

This study provided novel bigheaded carp and paddlefish presence and passage information related to environmental, hydraulic, and lock operations at UMR dams. Additionally, these results can have application to inform paddlefish and bigheaded carp management decisions at dams similar in operation and construction. Although this study adequately captured behaviors and passages of native and invasive species at a pinch-point dam in the UMR, additional considerations should be explored for future studies. This study had a low frequency of bigheaded carp residency events and passages documented from 2017 to 2019. We believe the low detection rates were a function of too few active bigheaded carp tags in the vicinity of LD 15. Deploying additional acoustically tagged bigheaded carp around LD 15 may elicit more challenges and passages at the dam, giving a more robust dataset for analyses. Additionally, future studies should consider incorporating a diversity of native, migratory species with unique sensory capabilities and life histories that would allow researchers to better understand how different species respond to different types of deterrents.

The increasing frequency of flood events in the UMR may limit the effectiveness of deterrents at locations where fish are able to complete upstream passage through the dam gates during periods of elevated discharge. In the past 10 years, the UMR has experienced six major floods (river stage exceeding flood stage at navigation dams) fueled by a combination of spring rainfall and snowmelt (*U.S. Army Corps of Engineers, 2020*). Current flood events last longer, are less predictable, and occur more frequently than those occurring in the 20th century. Over the past three years, this study has observed the number of open-river days at dams steadily rising (*i.e.,* LD 15 open-river days ranged between four to 108 days between 2017 to 2019 shown in Table S1; *Bouska, 2021*). The majority of upstream passages through the dam gates occurred during low hydraulic-head periods, typically co-occurring with major flooding. Increased opportunity for native fish passage is generally the goal for river managers, but the prolonged open-river condition and potential for upstream passage of bigheaded carp through the dam gates might create challenges for managers to limit the upstream expansion of invasive carp populations.

In years with infrequent open-river condition, a deterrent placed in the downstream lock approach may assist in meeting the management goal of reducing upstream passage of bigheaded carps. A combination of containment and control measures could provide the most effective tool for managing bigheaded carp in the UMR. The Upper Mississippi River Invasive Carp Team (UMRICT) is an interagency group across five states that is concerned with minimizing the impacts of bigheaded carp in the UMR (*Jackson & Runstrom, 2018*). Commercial harvest programs, funded through the UMRICT, are aimed at capturing and removing bigheaded carp in the UMR to prevent establishment of incipient populations

(*Jackson & Runstrom, 2018*). Bigheaded carp removal programs at least temporarily reduce populations and may help alleviate the pressure and number of challenge events invasive species elicit at dams. Fish deterrent technologies at pinch-point dams, paired with removal programs, could assist in preventing or reducing the upstream expansion of bigheaded carp in the UMR. Expanding upon information on fish passage at additional pinch-point dams for deterrent deployment, such as LD 14, could be useful if the reproductive front moves upstream past LD 15 (*Zielinski & Sorensen, 2021*). Lock and Dam 14 is less frequently at open-river condition than LD 15 and could potentially limit bigheaded carp passages through the dam gates during non-flood events. Bigheaded carp expansion beyond LD 14 is a great concern as the next potential pinch point is 145 rkm upstream at LD 11 (*Vallazza et al., 2021*). Understanding fish behavior at these dams is a critical information need for river managers as they evaluate potential tools or technologies that may assist in slowing or ceasing the upstream expansion of bigheaded carp in the UMR.

## ACKNOWLEDGEMENTS

We thank the U.S. Army Corp of Engineers-Lock Performance Management System for providing the Lock Queue Reports. We thank the graduate and undergraduate students Charmayne Anderson, Sabina Berry, Chelsea Center, Jehnsen Lebsock, Tyler Thomsen, Madeline Tomczak, Jesse Williams, Zachary Witzel, Madeline Davis, Lexie Froidcoeur, Fabian Guijosa, and Nick Vozza from Western Illinois University and Amanda Milde, Kyle Mosel, Bill Lamoreux, and Mark Fritts from USGS and USFWS for assistance in the field. We thank our commercial fishermen Shawn Price, Nick Dickau, and Charlie Gilpin for help collecting paddlefish for this study. We thank the reviewers for the helpful comments that strengthened this paper. Any use of trade, firm, or product names is for descriptive purposes only and does not imply endorsement by the U.S. Government.

### Funding

This work was supported by the U.S. Geological Survey Ecosystems Mission Area Invasive Species Program, U.S. Fish and Wildlife Service in support of Upper Mississippi River Invasive Carp Team priorities, and the Illinois Department of Natural Resources. The funders had no role in study design, data collection and analysis, decision to publish, or preparation of the manuscript.

### Grant Disclosures

The following grant information was disclosed by the authors:
The U.S. Geological Survey Ecosystems Mission Area Invasive Species Program.
U.S. Fish and Wildlife Service.
The Illinois Department of Natural Resources.

### Competing Interests

The authors declare there are no competing interests.

## Author Contributions

- Dominique D. Turney conceived and designed the experiments, performed the experiments, analyzed the data, prepared figures and/or tables, authored or reviewed drafts of the article, and approved the final draft.
- Andrea K. Fritts conceived and designed the experiments, analyzed the data, prepared figures and/or tables, authored or reviewed drafts of the article, and approved the final draft.
- Brent C. Knights conceived and designed the experiments, authored or reviewed drafts of the article, and approved the final draft.
- Jon M. Vallazza conceived and designed the experiments, analyzed the data, authored or reviewed drafts of the article, and approved the final draft.
- Douglas S. Appel conceived and designed the experiments, performed the experiments, prepared figures and/or tables, authored or reviewed drafts of the article, and approved the final draft.
- James T. Lamer conceived and designed the experiments, analyzed the data, authored or reviewed drafts of the article, and approved the final draft.

## Animal Ethics

The following information was supplied relating to ethical approvals (i.e., approving body and any reference numbers):

Western Illinois University IACUC committee.

## Data Availability

The data are available through ScienceBase: Fritts, A.K., Turney, D.D., Lamer, J.T., Knights, B.C., Vallazza, J.M., and Appel, D.S., 2022, 2017-2019 Telemetry data for invasive carp and paddlefish surrounding Lock and Dam 15 in the Upper Mississippi River Basin: U.S. Geological Survey data release, https://doi.org/10.5066/P9CHJ8OG.

## Supplemental Information

Supplemental information for this article can be found online at http://dx.doi.org/10.7717/peerj.13822#supplemental-information.

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
