# Peer review of "Hydrological and lock operation conditions associated with paddlefish and bigheaded carp dam passage on a large and small scale in the Upper Mississippi River (Pools 14–18)"

_PeerJ, doi:10.7717/peerj.13822_

## Round 0.1 · original submission · Minor Revisions

· Academic Editor

Minor Revisions

Both reviewers suggested the manuscript may benefit from additional clarity in places; these include additional discussion about the limitations of the dataset, and including additional citations in a few places.

Reviewer 1 ·

Basic reporting

In this manuscript, the authors present the findings of an acoustic telemetry study tracking movement of paddlefish, silver carp, and bighead carp through the Mississippi River Lock and Dam 14-18. The field study encompasses a large spatial region and provides insights into fish movement through Mississippi River Lock and Dams that have not been fully described in the past. While the sample sizes of silver and bighead carp are severely limited, the methodology and statistical analysis follow current best practices. The authors found that passage of all three species was closely tied to hydraulic head at the lock and dams among other covariates including lock usage, and water temperature. Overall, the manuscript is generally well written and methods adequately described.

Experimental design

The only major concern / question I have regarding the interpretation of the results is the reliance on hydraulic head as the sole covariate for hydraulic conditions at a lock and dam. While hydraulic head is an easily quantifiable variable, it is a poor descriptor of localized hydraulics that actually influence fish passage. For example, the same hydraulic head (difference between head- and tail-water levels) can be maintained by opening all spillway gates to the same, small opening versus opening only a few gates to a larger opening. Characteristic water velocity and turbulence downstream of each opening is going to vary drastically in spatial scale and impact on fish movement behavior. The authors need to acknowledge this imprecision in hydraulic head and address how this might limit insights from the current data set.

Validity of the findings

The findings are of general use for decisions regarding control strategies for invasive bigheaded carp, and are thus of interest for the journal audience.

Additional comments

Detailed comments are listed below with corresponding line numbers:
Line 86 – Insert “downstream” before “migration”
Line 93 – “Intentionally fragmented systems” is an awkward wording. I believe the authors mean to say “barriers” can prevent the spread of nonnative species.
Line 95 – While seasonal barriers are used to control sea lamprey in the Great Lakes, they are by no means “effective” at a broad scale as the migratory period of many native fish overlap substantially with sea lamprey migrations, as described by Velez-Espino et al. (2011).
Line 99 – Add Rahel and McLaughlin, 2018 here.
Line 103 – Delete “main thalweg”.
Line 106 – Replace “current” with “water”, replace “gates” with “spillway, and insert “gate” before “controlled”.
Line 107 – Lock and Dam #1 can’t be in open-river conditions due to the spillway type, and Lock and Dam 5 and 11 are also very similar in open-river occurrence.
Line 108 – Add reference to Wilcox et al. 2004.
Line 138 – Work by Zielinski et al. (2018) and Finger et al. (2019) have expanded on this possibility greatly.
Line 139-140 – This sentence is awkward, consider revision.
Line 188 – Are the same tag tpyes used for paddlefish and silver and bighead carp?
Line 216 – How representative is a 1-m depth to where paddlefish swim?
Line 234 – Why is hydraulic head used in the statistical analysis versus discharge or gate opening (see comment above)?
Line 251-255 – Why aren’t lock usage used in the TTE analysis but is used in later analyses?
Line 357-359 – Was any range testing done with a barge moving through the array? Presumably, the boats will cause line-of-sight issues and noise that may reduce tag detection.
Line 388 – 397 – What about bigheaded carp?
Line 417-420 – It might be useful for the authors to expand why up- and down-stream tows were analyzed separately. Is there something about lock operation in either scenario that differs and would impact the ability for fish to pass?
Line 493 – What conditions were present during the upstream passages that occurred outside of open-river?
Line 522 – While velocities through the partially closed gates may be high immediately under the gates, the spatial extent can be limited and create recirculation zones at different depths in the water column that could permit fish to nearly reach the gate without experiencing localized regions of high velocities. As part of this study, the authors have little to no quantified data on the flow conditions in or near the gates and thus discussion on specific flow conditions influencing fish movement is purely conjecture at this point. The flow conditions in and around the spillway gates are predictably very complex and citing data like “velocities as low as 0.6 m/s” from Wilcox (1999) simply perpetuates our general lack of understanding on the spatiotemporal hydraulic characteristics.
Line 526 – Morphological differences from bigheaded carp or interindividual differences? Please be more specific.
Line 524 – The reference to Wilcox (1999) is misleading at best. The value of “0.6 m/s” was not recorded, but was estimated from a physical model study completed by Markussen & Wilhelms (1987). The region of flow where 0.6 m/s was also spatially limited and not representative of the broader heterogeneity of the flow field through a spillway gate.
Markussen, J.V., Wilhelms, S.C., 1987. Scour Protection for Lock and Dams 2-10, Upper Mississippi River. Hydraulic Model Investigation. Technical Report HL-87-4. US Army Engineer Waterways Experiment Station.
Line 541 – This statement is counter intuitive as the highest localized velocities are expected to occur during high hydraulic head, which occurs during low discharge.
Line 608 – Large metal objects like the miter gates?
Line 621-626 – These recommendations related to acoustic deterrents are a bit outside the scope of this study and are not necessary. There are substantial lines of research already directed toward acoustic deterrents that are more current than Lovell et al. (2005 & 2006).
Line 650 – This exact idea has already been proposed by Zielinski and Sorensen (2021).
Zielinski, D. P. and Sorensen, P. W. (2021) Numeric simulation demonstrates that the upstream movement of invasive bigheaded carp can be blocked at sets of Mississippi River locks-and-dams using a combination of optimized spillway gate operations, lock deterrents, and carp removal. Fishes, 6(10), https://doi.org/10.3390/fishes6020010.

·

Basic reporting

1. Definition of Terms
Lines 56, 67, 68, 86, 103-104, 136, 332, 467, 469, 471, 480, 485, 493, 517.
Comment: The terminology is confusing, 1) the definition of open-river needs improvement, because technically there isn’t a “main thalweg spillway gates” part of the river, and 2) the generic use of “spillway passage” can refer to two very different parts of a dam; the adjustable spillway gates used to control water flow in the main channel (i.e. tainter and roller gates), and the fixed-crest spillway designed to release surplus flood water.
Recommendation: Change all parts of the document that refer to “spillway passage” to reflect that fish passed through the adjustable spillway gate openings in the main channel when the gates were positioned in the Open River condition. A generic discussion of terms could be helpful:

• Open River - occurs when the dam gates are lifted above the water, passing unobstructed flow through the adjustable spillway gates. During open river the head and tail water surface elevations of the river are nearly equal, essentially following the natural slope of the river.
• Spillway –Seventeen of the UMR navigation dams have fixed-crest spillways in the earthen dike sections of the dams, which are designed to pass flow during periods of high discharge. Most of these spillways have an ogee crest. Lock and Dam 15 is the only dam within the study area that does not have a fixed crest spillway. All water flows though the roller gates at L&D 15 and through the hydropower dams in Sylvan Slough.
• Thalweg - The thalweg is the deepest cross-sectional area of a river. At UMR dams the thalweg is technically in either the main lock or the auxiliary lock chambers. Most UMR dams are designed so that the deepest part of the river allows for navigation traffic and emergency dewatering of the upstream pool.

Experimental design

No comment

Validity of the findings

2. Un-cited Literature - Several additional papers should be considered in the discussion

Lines 138-142 and 641-660
Comment: Gate manipulations have been considered to block fish movements in addition to those mentioned on line 142. I was expecting to see some discussion/comparisons with the similar fish movement study conducted at L&D 2 (even though the species are different).

Recommendation: If not further discussed, at a minimum, Finger et al (2020) should be referenced and gate manipulations added to the list of deterrents on line 142.

Finger, Jean Sebastien, Andrew Thomas Riesgraf, Daniel Patrick Zielinski, Peter William Sorensen. 2020. Monitoring upstream fish passage through a Mississippi River lock and dam reveals species differences in lock chamber usage and supports a fish passage model which describes velocity‐dependent passage through spillway gates. River Research and Applications 36:36–46.

Lines 549 – 556
Comment: Add a discussion regarding Maynord (2005) observations to explain the hydraulic differences between upbound and downbound lockage.

Recommendation: Add information regarding; “The average upbound unloaded tow from the field study passed 49 percent of the water volume of the lock through the wheels. The average downbound loaded tow from the field study passed 228 percent of the water volume of the lock through the wheels (Maynord 2005).” plus a discussion on how this information may have impacted fish passage in the lock.

Maynord, Stephen T. 2005. Flow through the wheels of towboats in 600 ft locks on the UMR-IWW. Upper Mississippi River. U.S. Army Corps of Engineers. Upper Mississippi River - Illinois Waterway System Navigation Study ENV Report 59. https://www.mvr.usace.army.mil/Missions/Navigation/NESP/Related-Documents-Info/FileId/292281/

Lines 557-563
Comment: Seasonal abundance of fish has been the focus of other studies on the UMR.

Recommendation: Add discussion regarding Johnson et al (2005) observations to bolster conclusions. “High fish densities in the lock during the spring, summer, and fall (mid-April through November) and low densities during the winter reflect the abundance of fish in the main channel during these periods (Johnson et al 2005).” This information seems to corroborate the study findings and should be cited.

Johnson, Brian L., Thomas M. Keevin, Eric A. Laux, and Thixton B. Miller, Donald J. Degan, David J. Schaeffer. 2005. Seasonal Fish Densities in the Lock Chamber at Lock and Dam 25. Upper Mississippi River. U.S. Army Corps of Engineers. Upper Mississippi River - Illinois Waterway System Navigation Study ENV Report 57 https://www.mvr.usace.army.mil/Missions/Navigation/NESP/Related-Documents-Info/FileId/292279/

Lines 557 – 563
Comment: Seasonal differences in residency time have been observed previously. Add citation and/or discussion regarding Keevin et al (2005) findings for comparison.

Recommendation: Add discussion regarding Keevin et al (2005) observations to bolster conclusions.
“High August mortality reflects the large numbers of fish in the lock; high December mortality possibly reflects the inability of the fish in the lock to avoid entrainment because of reduced swimming capabilities.”

“…mortality was related to water temperature, depth of water in the lock, horsepower of the tow, number of barges, and time of year.”

Keevin, Thomas M., Brian L. Johnson, Eric A. Laux, Thixton B. Miller, Kevin P. Slattery, David J. Schaeffer. 2005. Adult fish mortality during lockage of commercial navigation traffic at Lock and Dam 25, Upper Mississippi River. U.S. Army Corps of Engineers. Upper Mississippi River - Illinois Waterway System Navigation Study ENV Report 58.

Additional comments

3. Clarifications
Line 82.
Comment: The statement “Navigation dams are known to impede fish passage in lotic systems”. Though true, this statement is restrictive. All dams, not just navigation dams, impede fish passage to some degree.
Recommendation: Remove the word “Navigation”.

Lines 165 - 166
Comment: The sentence “Lock and Dam 15 has nine non-overflow roller gates and two 33-m overflow roller gates, a main lock, and a functioning auxiliary lock.” is confusing and incomplete.
Recommendation: Re-word the sentence and add a discussion of the hydropower dams. Lock and Dam 15 has 11 roller gates and two locks, a main and an auxiliary lock. It also has hydropower dams located in a secondary channel on the east side of Arsenal Island. These dams are located approximately 1.5 miles upstream of Lock and Dam 15 on either side of Sylvan Island in Sylvan Slough.

Lines 504-507
Comment: The sentence “These locks and dams have been identified as potential targets for deterrent technologies (Upper Mississippi River Asian Carp Partnership 2018) and implementation of effective deterrents at these dams could limit additional bigheaded carp range expansion in the UMR.” should be reduced.
Recommendation: Simplify the sentence to something like; They have been identified as potential locations for fish deterrent technologies to limit bigheaded carp range expansion in the UMR (Upper Mississippi River Asian Carp Partnership 2018).


4. Typographical comments
Lines 644-645.
Comment: UMRACT has recently changes its name to UMRICT.
Recommendation: Change sentence to update the name change to Upper Mississippi River Invasive Carp Team, formerly known as the Upper Mississippi River Asian Carp Team.

Add volume and page numbers to line 713 and 905:
711 Camacho CA, Sullivan CJ, Weber MJ, Pierce CL. 2020. Invasive carp reproduction
712 phenology in tributaries of the Upper Mississippi River. North American Journal of
713 Fisheries Management DOI: 10.1002/nafm.10499.

903 Vallazza JM, Mosel KJ, Reineke DM, Runstrom AL, Larson JH, Knights BC. 2021. Timing and
904 hydrological conditions associated with bigheaded carp movement past navigation dams
905 on the upper Mississippi River. Biological Invasions DOI:
906 https://doi.org/10.1007/s10530-021-02583-8.

Remove parenthesis around 2018
887 Upper Mississippi River Asian Carp Partnership (2018). Potential use of deterrents to manage
888 Asian carp in the upper Mississippi River basin. http://www.micrarivers.org/wp-
889 content/uploads/2019/08/Potential-Use-of-Deterrents_Final.pdf

---

## Round 0.2 · accepted · Accept

· Academic Editor

Accept

I appreciate your attention in clearly addressing the reviewers' comments, and think the submitted manuscript has been improved.